# Quantum-Chemical Quasi-Docking for Molecular Dynamics Calculations

**DOI:** 10.3390/nano12020274

**Published:** 2022-01-15

**Authors:** Alexey Sulimov, Danil Kutov, Ivan Ilin, Vladimir Sulimov

**Affiliations:** 1Dimonta, Ltd., 117186 Moscow, Russia; sulimovv@mail.ru (A.S.); ilin@dimonta.com (I.I.); 2Research Computer Center, Lomonosov Moscow State University, 119992 Moscow, Russia

**Keywords:** docking, quantum chemistry, supercomputing, drug design, inhibitor, target protein, global energy minimum, force field, continuum solvent model

## Abstract

The quantum quasi-docking procedure is used to compare the docking accuracies of two quantum-chemical semiempirical methods, namely, PM6-D3H4X and PM7. Quantum quasi-docking is an approximation to quantum docking. In quantum docking, it is necessary to search directly for the global minimum of the energy of the protein-ligand complex calculated by the quantum-chemical method. In quantum quasi-docking, firstly, we look for a wide spectrum of low-energy minima, calculated using the MMFF94 force field, and secondly, we recalculate the energies of all these minima using the quantum-chemical method, and among these recalculated energies we determine the lowest energy and the corresponding ligand position. Both PM6-D3H4X and PM7 are novel methods that describe well-dispersion interactions, hydrogen and halogen bonds. The PM6-D3H4X and PM7 methods are used with the COSMO implicit solvent model as it is implemented in the MOPAC program. The comparison is made for 25 high quality protein-ligand complexes. Firstly, the docking positioning accuracies have been compared, and we demonstrated that PM7+COSMO provides better positioning accuracy than PM6-D3H4X. Secondly, we found that PM7+COSMO demonstrates a much higher correlation between the calculated and measured protein–ligand binding enthalpies than PM6-D3H4X. For future quantum docking PM7+COSMO is preferable, but the COSMO model must be improved.

## 1. Introduction

The Coronavirus (COVID-19) pandemic [1], caused by coronavirus SARS-CoV-2 [2], has highlighted the urgent need for docking programs. These programs are used to search for compounds for which molecules (ligands) can selectively bind to the active site of a therapeutic target protein. The latter is responsible for the progression of the disease. If the protein is inhibited, that is, blocked by ligand binding, the progression of the disease is suppressed or even stopped. The docking programs find the position of the ligand in the active site of the target protein and estimate the free energy of protein-ligand binding. Ligands, i.e., inhibitors, with high binding energy can become active components of new drugs, and the higher the binding energy, the more effective drug creation becomes using such an inhibitor, since a lower concentration of the inhibitor leads to a therapeutic effect. Docking programs are used at the initial stage of the drug design pipeline to screen large databases of organic compounds containing many thousands or even millions of ligands. The selected ligands with the highest estimated binding energies are tested experimentally in vitro, e.g., in a test system with the target protein and its fluorogenic substrate, to confirm their ability to inhibit the target protein. Typically, experimental confirmation of inhibitory activity is obtained only for a very low percentage of tested compounds due to insufficient docking accuracy. Molecular dynamics (MD) is used to increase the proportion of experimentally proven inhibitors. However, molecular dynamics methods, for example, the Free Energy Perturbation method (FEP), can be applied only to a very limited number (less than several tens) of ligands selected by docking, due to the number of computational resources required for MD methods. The chemical structures of experimentally proven inhibitors are further optimized to increase their binding affinity. This optimization is carried out either purely experimentally, by the chemical synthesis of new compounds with subsequent experimental testing of their inhibitory activity, or by preliminary performing more accurate calculations of the free energy of protein-ligand binding using molecular dynamics. An excellent illustration of such an approach is presented in two recent works [3,4]. In the first one [3], approximately 2000 ligands of approved drugs are screened by four docking programs to identify inhibitors of the main protease (M^pro^) of the SARS-CoV-2 coronavirus responsible for COVID-19. Using a consensus approach 42 best candidates are selected on the basis of the docking results, which are subsequently inspected by molecular dynamics modeling and only 17 ligands are chosen for experimental trials. Only two of them show noticeable inhibition with IC_50_ < 10 μM, where IC_50_ is a measure of inhibitory activity corresponding to the concentration of an inhibitor that suppresses the activity of the target protein by 50%. In the subsequent work [4], the chemical structure of a very weak inhibitor, parampatel (IC_50_ ca. 100 μM), found in [3], is optimized using FEP and the several resulting synthesized compounds demonstrate IC_50_ ca. 20 nM, and some of them suppress replication of live SARS-CoV-2 in cell culture at the micromolar level. On the other hand, purely experimental screening of 10,755 compounds [5], approved drugs and drug candidates, results in the determination of 10 inhibitors of M^pro^ with IC_50_ in the range from 0.26 to 13.00 μM and only two of them suppress the replication of live SARS-CoV-2 in the cell culture. Similar experimental screening is described in [6], wherein, among the 8702 drug candidates, only 7 new micromolar inhibitors of M^pro^ are found, only two of which suppress the replication of the virus. These examples show that docking, together with molecular dynamics, can greatly increase the effectiveness of new inhibitors development.

Given that a very large percentage of inhibitors are typically filtered out in subsequent expensive and time-consuming pre-clinical and clinical trials, improved docking accuracy is imperative to improve the efficiency of the entire drug development pipeline. Thus, the two molecular modeling methods, docking and molecular dynamics, complement each other in drug development, and the accuracy of the former determines the effectiveness of the latter.

This work attempts to investigate the docking accuracy. At present, there are several tens of docking programs. Comprehensive reviews on docking can be found in recent publications [7,8,9] and references therein. The performance of most of docking programs is based on the docking paradigm that assumes that the position of the bound ligand in the active site of the target protein is near the global minimum of the energy of the protein–ligand complex. Thus, the problem of ligand positioning in the active site of the target protein is reduced to the problem of global minimization on a multidimensional complicated, i.e., with many local minima, energy surface. The dimension of the surface is defined by the number of variable degrees of freedom, that can be up to 10–20 for a flexible ligand and a rigid protein. The solution to this problem in itself is extremely difficult but there are many other factors that reduce the docking accuracy [10]. One of them is force fields, sets of classical interatomic potentials, which are used to calculate the energy of the protein-ligand complex in almost all docking programs. However, it has been well known for almost 100 years that interatomic interactions in molecules and intermolecular interactions must be described by quantum chemistry, that is, quantum mechanics applied to molecules. Unfortunately, the size of the protein-ligand complexes (several thousand atoms) is too large for the global optimization of the energy of the protein-ligand complex calculated by the methods of quantum chemistry. Therefore, the docking is carried out by force fields. There are no quantum-chemical docking programs as of yet. In some publications (see examples in [9]), quantum-chemical docking is declared but actually quantum-chemical methods are used in post-processing after classical docking to improve estimates of protein-ligand binding energy. This state of affairs is also true for molecular dynamics applied to protein-ligand complexes—there are no quantum molecular dynamics programs for such large molecular systems.

An approach to performing quantum-chemical docking, quasi-docking, has been recently proposed in [11,12,13]. First, a wide spectrum of low energy minima of the protein-ligand complex is found using a force field. Then, energies of these minima are recalculated by the quantum-chemical method, the global energy minimum and the corresponding position of the ligand in the active site of the target proteins are determined and the task of quantum docking is completed. This procedure is based on the assumption that the position of the ligand corresponding to the global minimum of the quantum-chemical energy of the protein-ligand complex is among the positions of the ligand corresponding to the energy minima found at the first stage of the quasi-docking procedure. In quantum docking (it is not realized yet), we must directly carry out the search for the global minimum of the energy of the protein-ligand complex calculated by the quantum-chemical method. Practically, two recently developed quantum-chemical semiempirical methods, PM6-D3H4X [14,15] and PM7 [16], can be used in this procedure. These two methods excel all previous semiempirical methods in their accuracy of description of dispersion interactions, hydrogen and halogen bonds. These methods are compared with each other on different sets of low molecular weight complexes in [17], and PM6-D3H4X gives the best results. However, in [12], the PM7 method demonstrates slightly better ligand positioning accuracy than PM6-D3H4X in quasi-docking. In [13], PM7 demonstrates not only high positioning accuracy, but also a high correlation between the calculated and measured values of the enthalpy of protein-ligand binding. In this paper, we compare the accuracy of PM7 and PM6-D3H4X methods in the quasi-docking procedure and PM7 gives the best results.

## 2. Materials and Methods

### 2.1. Main Methods and Software

The first step of the quasi-docking procedure is carried out using the supercomputer generalized docking program FLM [18,19,20,21], where the abbreviation FLM means “Find Local Minima”. FLM searches for and collects low energy minima of a protein–ligand complex by randomly throwing a flexible ligand into the active site of a rigid target protein and optimizing the energy of the protein-ligand complex using the L-BFGS method [22,23] by varying Cartesian coordinates of all atoms of the ligand. The initial positions of the ligand in the active site are generated by random continuous translations and rotations of the ligand as a rigid body and random continuous changes in torsions of the ligand, that is, internal rotations around ordinary chemical bonds. The initial pose of the ligand is rejected when ligand atoms clash with each other or with the protein atoms. The only restriction on the position of the ligand is that the geometrical center of the ligand must be located within a given sphere (the default radius is 8 Å) covering the active site. In this process, the uniqueness of the local energy minima is accounted for, and the unique low energy minima are collected in a special pool of a given size. The uniqueness of the minima is determined by calculating the standard deviation (RMSD) between each minimum found and all the minima in the pool. During local optimization, the energy of the protein-ligand complex in each conformation is calculated within the MMFF94 force field [24] (the initial published version of the Merck molecular force field, MMFF) without any corrections or fitting parameters. The energy calculation can be done with or without solvent. When the solvent is taken into account, the continual Surface Generalized Born (SGB) solvent model [25] is used. In [21], the possibility of finding the global energy minimum of the protein-ligand complex calculated by the PM7 method with the COSMO solvent model using the FLM program without a solvent at the first stage of quasi-docking was shown. Therefore, in this study, the FLM program is used without solvent at the first step of quasi-docking.

FLM is a parallel program that uses the Message Passing Interface (MPI), and runs on a given number of computing cores for as long as possible, depending on the available supercomputer resources. Each local optimization runs on a single computing core, resulting in linear scaling of FLM performance as the number of cores increases. The size of the pool of low energy minima is equal to 8192 by default and it contains the global energy minimum and every successive energy minimum above it. The completeness of the unique low energy minima spectrum, when there are no missed energy minima, is an important feature of the FLM performance. However, for some complexes (see Table 1), we were forced to preserve much more energy minima, because among the 8192 lowest energy minima found at the first stage of quasi-docking, none corresponded to the ligand position close to the position of the crystallized native ligand. The time of FLM performance was selected to be sufficient (for a given protein-ligand complex) to reach a saturation of a wide spectrum of unique low energy minima. Here, the term “saturation” means that a given number of unique low energy minima is found, and further continuation of the search for low energy minima does not reveal any additional minimum, the energy of which is lower than the energy of any minimum from the found spectrum. A comprehensive statistical analysis of the convergence of the spectrum of unique low energy minima, which were determined by the FLM program, were provided in our previous studies [18,20,21,26].

FLM determines all low energy minima of one protein–ligand complex for a large running time, depending on the numbers of ligand atoms and torsions and can be as long as 27,000 CPU × hours for a ligand with 74 atoms and 19 torsions and as small as 1000 CPU × hours for a ligand with 20 atoms and 2 torsions [20].

At the second step of the quasi-docking procedure, energies of all unique local energy minima of the protein-ligand complex found using the MMFF94 force field at the first step of quasi-docking are recalculated by the MOPAC (the name is the abbreviation of “Molecular Orbital PACkage”) program [27] using two quantum-chemical methods, PM6-D3H4X and PM7 with the COSMO solvent model [28]. The name COSMO is an abbreviation of the “Conductor-like Screening Model”. This solvent model describes the polar part of the solute-solvent interaction approximating the aqueous solvent around the molecule as a homogeneous medium with an infinitely high dielectric constant, then a correction is made to the polarization energy by the finite value of the dielectric constant of water (78,4 at room temperature). We previously found that the energy recalculation using the PM7 method without COSMO led to a reduced positioning accuracy [18]. We compared quasi-docking procedure PM7 with the COSMO solvent model (for the sake of brevity, we will refer to such calculations as PM7+COSMO), parameterized without non-polar solute-solvent term, and PM7 with the new COSMO2 solvent model [29], parameterized with non-polar solute-solvent contribution using the SASA approximation [13]. SASA is an abbreviation of “Solvent Accessible Surface Area”. We found that PM7 in conjunction with COSMO2 showed worse results in ligand positioning and also in the correlation of the calculated and measured values of the binding enthalpy. This is why we use the COSMO model here to compare the PM7 and PM6-D3H4X methods in the quasi-socking procedure. Improving the COSMO solvent model by reparameterizing it for the PM7 and PM6-D3H4X methods and incorporating a more elaborated non-polar solute-solvent interaction model, than the popular but oversimplified SASA approximation, e.g., as in [30] is beyond the scope of this article. Recalculation of energies of protein-ligand complexes are performed using the MOZYME module [31] implemented in MOPAC, which allows much faster calculation of systems containing several thousand atoms using localized molecular orbitals. In this recalculation, no local optimization of the energy of the protein-ligand complex is performed, i.e., the keyword 1SCF is used in MOPAC calculations. The ligand position corresponding to the minimum with lowest value of the quantum-chemical energy calculated with the COSMO solvent model is considered as the global minimum (GM) of the energy of the protein–ligand complex, which is calculated using the quantum-chemical method. Furthermore, we take into account that the solvent is important for good positioning accuracy [11,12,18]. Verification of the docking paradigm feasibility is carried out by calculating the RMSD between the position of the ligand in the GM and the position of the ligand in the crystallized complex. We assume that the docking paradigm is fulfilled if RMSD < 2 Å. Previously,, it was shown that the FLM program identified a deeper global energy minimum, and calculated the MMFF94 force field at a lower energy minimum than that found by the SOL classical docking program [32] (see Table 2 in [20]). In addition, the entire quasi-docking procedure with the PM7+COSMO method finds a global energy minimum close to the position of the crystallized native ligand for more test complexes than SOL does [13]. The positioning accuracy of quasi-docking is better than one of the classical docking with the classical force field [13,20]. The protein–ligand binding enthalpy is calculated only for such complexes with good positioning accuracy of the quasi-docking procedure.

The protein-ligand binding enthalpy is calculated as follows:ΔH_bind_ = E_1_(PL) − E_1_(P) − E_1_(L),(1)
where E_1_(*X*) is the energy of the global minimum of *X* = PL (protein-ligand complex), P (unbound protein) or L (unbound ligand). As a first approximation it is possible to neglect contributions from higher energy minima and vibrations [18].

The E_1_(PL) energy is calculated as follows. The energy of the protein–ligand complex is optimized by varying the Cartesian coordinates of all ligand atoms using either PM6-D3H4X or PM7 without solvent from the ligand position corresponding to GM found in the quasi-docking procedure. Then, the energy of the protein-ligand complex at the achieved minimum is recalculated by the corresponding quantum-chemical method using the COSMO solvent model. This energy of the protein-ligand complex is E_1_(PL). The E_1_(P) energy of the unbound protein is calculated for the fixed conformation of the protein used for docking at the first step of the quasi-docking procedure, and no local optimization was carried out. The E_1_(L) energy was calculated using a procedure similar to quasi-docking applied to the unbound ligand, but at the second step, the ligand quantum-chemical energies calculated by PM6D3H4X or PM7 without solvent are optimized locally by varying Cartesian coordinates of all ligand atoms from initial ligand conformation corresponding to each energy minimum found at the first step using the MMFF94 force field. Finally, energies of all the obtained minima are recalculated using either PM6-D3H4X or PM7with the COSMO solvent model and the E_1_(L) value is determined as the lowest PM6-D3H4X+COSMO or PM7+COSMO energy. The E_1_(*X*) value, *X* = PL, P and L, is the heat of formation in the output of MOPAC.

### 2.2. Test Set of Protein-Ligand Complexes

A test set of 25 protein-ligand complexes containing 12 different proteins and ligands of various size and flexibility with experimentally measured thermodynamic binding characteristics was selected using the following criteria [13]. The structures of protein-ligand complexes taken from Protein Data Bank [33] must be of a high quality, with a resolution of better than 2.0 Å, atoms or amino acid residues in the active site of the protein will not be missed, no metal ions and cofactors crystallized in the active site of the protein, and experimentally measured binding characteristics. Firstly, the protein-ligand binding enthalpy, must be known. The latter allows to put aside the problem of the calculation of entropy contribution to the protein-ligand binding free energy. Protein and ligand protonation are achieved using the Aplite [34] and Avogadro [35] programs, respectively, for pH = 7.4. Proteins are protonated at a pH of 7.4 due to the fact that the natural “working” molecular environment of proteins listed in the test set possesses this pH value. Such an approach to selecting protonation pH is justified by the medicinal chemistry common practice consideration of modeling conditions, which rely upon the natural context of a biomolecule in the body. There is a specific feature of protonation of the native ligand from the 2IKO complex. The PoseView plugin used on the Protein Data Bank website (www.rcsb.org, accessed on 1 January 2022) presents this ligand as a neutral molecule. However, a basic characteristic of the diaminopyrimidine fragment of the ligand allows it to accept a proton from a neighboring cluster of aspartic acids and acquire a positive charge upon binding. In the crystallized 2IKO complex, diaminopyrimidine is directed towards this cluster, and this also confirms the need to use the charge of the ligand +1, which was done in our calculations.

The quality of the complexes is also reviewed according to the following criterion, which reveal the ability of the MMFF94 force field to correctly describe the protein–ligand interaction at the first step of the quasi-docking procedure. The MMFF94 energy of each protein-ligand complex is optimized by varying Cartesian coordinates of all ligand atoms starting from the crystallized ligand position. If optimized and crystallized ligand positions differ by RMSD < 2 Å, the complex is included in the test set. The main characteristics of the test complexes are presented in Table 1.

## 3. Results and Discussions

The task of the quasi-docking procedure is to position the ligand in the active site of the target protein. In accordance with the docking paradigm, the position of the ligand in the active site of the protein corresponds to the global minimum (GM) of the energy of the protein-ligand complex, which is calculated using the quantum-chemical method, and in our case, using either the PM6-D3H4X or the PM7 with the COSMO solvent model. This GM is found by the two-step quasi-docking procedure. Then, the ligand position corresponding to this GM is used for the calculation of the protein–ligand binding enthalpy.

### 3.1. Ligand Positioning

It is convenient to introduce the indices of the energy minima found using the quasi-docking procedure. These minima are ranked in ascending order of their energy calculated at the second step of quasi-docking. Each minimum is assigned its own integer index that is equal to its position number in this ranked list. The index of the global energy minimum is equal to 1. Some minima can correspond to ligand positions close to the position of the native ligand crystallized with the protein. We define this closeness as RMSD < 2 Å, where RMSD is calculated over heavy, i.e., not hydrogen, ligand atoms. There can be several such minima, and among them we denote the index of the minimum with lowest energy as INN, where INN means Index Near Native. If INN is 1, the docking paradigm is satisfied, that is, the position of the ligand in the protein corresponding to the global energy minimum of the protein-ligand complex is close to the position of the ligand crystallized with the protein. In Table 2, indices INN calculated by quasi-docking using either PM6-D3H4X or PM7 with the COSMO solvent are presented.

We see that quasi-docking by the PM6-D3H4X method with the COSMO solvent model cannot position native ligands using the docking paradigm for 11 of the 25 test complexes (INN > 1). On the other hand, quasi-docking by the PM7 method with COSMO fails to reproduce positions of native ligands for only 6 out of 25 complexes. So, positioning accuracy of quasi-docking by PM7+COSMO is much better than the accuracy of quasi-docking by PM6-D3H4X+COSMO. Comparison of the quasi-docking results obtained using PM7+COSMO (six cases INN > 1) and PM7+COSMO2 (eleven cases INN > 1) shows that the positioning accuracy of the former is better than the latter. The reasons for the inability of quasi-docking with PM7+COSMO to reproduce the crystallized poses of some native ligands are as follows [13].

First, there are symmetrical binding sites that allow ligands to bind in two alternative symmetric conformations. This problem arises when managing homodimers, for the 2PYM and 4LL3 complexes containing the HIV-1 protease. When the alternative symmetric conformation of the native ligand was retrieved from Protein Data Bank, the RMSD between the docked ligand position corresponding to the global energy minimum and the native ligand position was found to be equal to 1.66 Å, and we assigned INN = 1 to the quasi-docking results for the 2PYM complex. For the 4LL3 complex, the RMSD was found to be equal to 4.78 instead of 8.38 Å.

Second, the position of the ligand corresponding to the global energy minimum differs from the crystallized position of the native ligand only in the position of one ligand fragment, but this difference does not affect the ability of the bound ligand to block the catalytic triad of the protein. This was observed in the 2ZFS and 4CRC complexes, where the RMSD is slightly more than 2 Å.

The mismatch between the charge of the ligand during the simulation and the charge of the crystallized ligand can lead to bad quasi-docking accuracy. This is the case of the 5CSD complex. Its crystallization occurred at pH = 4.8, at which an arachidonic acid, a native ligand of 5CSD, exists in both (deprotonated and protonated) states since its pKa is 4.752. The hydrophobic nature of the binding site facilitates the binding of the deprotonated and neutral form of the ligand as compared to the protonated form. However, we studied the protonated form, which could affect the results.

### 3.2. Binding Enthalpy

It is difficult to expect a good accuracy of the calculated binding enthalpy if the positioning accuracy is unsatisfactory. Therefore, we calculated the protein-ligand binding enthalpy only for those complexes for which the docking paradigm is fulfilled and INN = 1. For some complexes presented in Table 1 (3KIV, 4CRC and 4CRD), the experimentally measured protein-ligand binding enthalpy ΔH_bind_ was unknown. We exclude the 2PYM and 2PYN complexes from the calculation of the binding enthalpy, because ΔH_exp_ > 0, and the process of ligand-protein binding is determined by the change in entropy upon binding. We excluded complexes 1J84, 2Z8E and 4P8V here, because their native carbohydrate ligands contain distorted pyranose rings in either bound or unbound ligands. The calculated protein-ligand binding enthalpy values ΔH_exp_ for the test complexes are presented in Table 3.

Table 3 shows that the bound state of the protein and the ligand is energetically more preferable than the unbound state for all complexes, that is, ΔH_bind_ < 0. The correlation coefficient between the measured ΔH_exp_ and calculated ΔH_bind_ values of the protein-ligand binding enthalpy for PM7+COSMO is equal to R = 0.74, which is much higher than for PM6-D3H4X+COSMO (R = 0.4). It should also be noted that the correlation coefficient between the values of the calculated and measured enthalpy of binding increases slightly from 0.74 to R = 0.77 for the PM7+COSMO method, if the energy of the ligand in the bound and unbound states is optimized with a solvent, i.e., using PM7+COSMO. As mentioned above, the values presented in Table 3 were obtained when the local energy optimization was carried out without a solvent (with PM7 or PM6-D3H4X), and the solvent was taken into account only to recalculate the energy for the ligand conformation corresponding to the minimum energy found without solvent.

Thus, the calculation of the energy using the PM7 method and the COSMO solvent model in the quasi-docking procedure provides a much better positioning accuracy and better correlation with the measured values of the protein-ligand binding enthalpy than corresponding values calculated using PM6-D3H4X and the COSMO model. However, the calculated by the PM7+COSMO method protein-ligand binding enthalpies are much more negative than the measured ones. Values of the protein-ligand binding enthalpy calculated by the PM6-D3H4X+COSMO method are much lower but the correlation with experimental values is bad (R = 0.4).

Higher values of the binding enthalpy were also obtained when calculating the interactions of ligands with parts of proteins containing only their active sites in [36], in which it is claimed that “…the total interaction energy between an active site and a ligand is never smaller than 50 kcal/mol”. Additional efforts are required to overcome this discrepancy. Perhaps the situation will be improved by additional features of the model.

For example, inclusion of the protein atoms neighbors of ligand atoms in the local optimization of the energy of the protein ligand complex. For the self-consistency, the energy of the unbound protein must be optimized also by varying positions of the same protein atoms. A preliminary study shows that the decrease of the energy of the unbound protein is higher that the decrease of the energy of the protein-ligand complex. Therefore, the binding enthalpy ΔH_bind_ calculated according to Equation (1) becomes less negative. An improvement is clearly required for the model of nonpolar interaction of a solute and solvent and its self-consistent implementation together with the COSMO model in the MOPAC program. However, more research is needed to verify this finding.

## 4. Discussion

Predicting the binding of a ligand to the target protein is a key step in drug development. Improving the reliability of this prediction leads to an increase in the efficiency of the entire drug development pipeline. Now, the prediction can be made on the basis of knowledge of the molecular mechanisms of many diseases, the corresponding therapeutic target proteins and their atomistic structures. Extensive experience in the development and application of docking programs to address this problem provides the basis for a roadmap to improve the efficiency of drug development using computer simulations. All existing docking programs use classical force fields, which in many cases have been further simplified [7,8,9]. However, the intermolecular interaction between the ligand and the target protein must be described by quantum mechanics using quantum-chemical methods. This is one of the most important factors limiting docking accuracy and docking efficiency in drug development. Some time ago, we proposed the quasi-docking procedure as the first step towards quantum docking. In using the phrase “quantum docking” we mean such a docking algorithm and the corresponding program that is able to search for the best position of the ligand in the protein based on the search for the global energy minimum of the protein-ligand complex, when the energy is calculated by the quantum chemical method. Quasi-docking using the PM7 quantum chemical method with the COSMO solvent model has already demonstrated much better positioning accuracy than docking accuracy with force fields [10,11,12]. We did not calculate the correlation between experimentally measured binding enthalpy values and values of binding enthalpy calculated using MM-GBSA/MM-PBSA methods. However, we investigated docking positioning accuracy of the MM-GBSA method in several of our articles [10,11,12], and we showed that docking positioning accuracy with the MM-GBSA method is much worse than the docking positioning accuracy of quasi-docking with PM7+COSMO. It does not make sense to calculate the binding enthalpy using the wrong ligand positions found using the MM-GBSA method.

The relatively new PM7 method describes intermolecular interactions much better than the old semiempirical AM1 and PM3 methods. The implementation of PM7 with the MOZYME module in the MOPAC program makes it possible to use PM7 with COSMO for the calculating protein-ligand complexes. However, there is another new quantum-chemical semiempirical method, PM6-D3H4X, which describes intermolecular interactions with the same degree of accuracy as PM7 or better [17].

In this study, we demonstrated, by quasi-docking, that PM7 with the COSMO model exhibits high accuracy of the ligand positioning in the protein, and this accuracy is higher than the positioning accuracy when PM6-D3H4X is used with the same COSMO model. The protein–ligand binding enthalpies calculated using the first method correlate much better with the experimental values than the values calculated using the second method. These results are obtained for a test set of 25 protein-ligand complexes in the frame of the limitations of the existing COSMO model implemented in the MOPAC program. We emphasize that no fitting parameters were used in these calculations, and the force field at the first stage and the PM7 (or PM6-D3H4X) method and the COSMO solvent model at the second stage of quasi-docking are the same as were created.

Evidently, the quasi-docking procedure cannot compete with the MM-GBSA method and with the most widespread “classic” docking programs [7,8,9] in terms of the speed of virtual screening of large databases containing many thousands and millions of ligands. Quasi-docking with the PM7+COSMO method can be useful in the hit-to-lead optimization phase in addition to molecular dynamics (MD) modeling. However, quasi-docking has an advantage, since it uses the quantum-chemical method, rather than classical force fields, as is done in MD simulation. In addition, quasi-docking does not use fitting parameters, as is done in “classic” docking programs [7,8,9]. Therefore, when using quasi-docking, the accuracy of calculations can be better controlled, hence the efficiency of using docking for drug development can be increased. It is clear that quasi-docking requires enormous computational resources, such as a future quantum docking program, but the computing power of supercomputers has grown so rapidly over the past 20 years that we assume that quantum docking will become a common tool in a few years.

## 5. Conclusions

We have shown that the PM7 quantum-chemical method with the COSMO solvent model exhibits a high accuracy of ligand positioning in the quasi-docking procedure by combining a classical force field and the quantum-chemical method. For 19 test protein-ligand complexes out of 25, quasi-docking found ligand positions corresponding to the global energy minima calculated using the PM7+COSMO method near (RMSD < 2 Å) the crystallized native ligand position in the protein. The high correlation (0.74) between the experimentally measured values of the protein-ligand binding enthalpy and those calculated with PM7+COSMO is demonstrated. Quasi-docking using the PM6-D3H4X quantum-chemical method with the same COSMO solvent demonstrates much worse positioning accuracy and the correlation between measured and calculated values of the protein-ligand binding enthalpy.

Although the quasi-docking positioning accuracy with the PM7+COSMO method and the correlation between the experimentally measured and calculated values of the protein-ligand binding enthalpy are high, the calculated values deviate strongly from the measured ones. To eliminate this discrepancy, it may be necessary to take into account the relaxation of the positions of the protein atoms near the bound ligand. Another source of the strong deviation calculated protein–ligand binding enthalpies from measured ones might be connected to imperfections of the COSMO model. Among the necessary improvements, we identify the reparameterization of the COSMO model for the PM7 and PM6-D3H4X methods by including a model for nonpolar solute-solvent interaction which is better than the oversimplified SASA approximation.

The obtained results show that the quantum docking program can be developed using the PM7+COSMO method for the calculation of the energy of a protein-ligand complex. Realizing quantum docking will improve the docking accuracy and increase efficiency of docking application in drug development. However, the realization of quantum docking needs the new docking algorithm which is much more effective than the commonly used genetic algorithm of the global energy optimization.

The quasi-docking procedure, which performs quantum-chemical positioning of the ligand in the active site of the target protein, can complement classical molecular dynamics in the computer-aided structural-based drug design, and both of them, together with classical docking, should significantly increase the effectiveness of existing drug development pipelines.

## Figures and Tables

**Table 1 nanomaterials-12-00274-t001:** Protein-ligand complexes selected for the test set and their characteristics: PDB ID is the identifier of the complex in Protein Data Bank, protein names, Res is the resolution of the complex, N_P_, N_L_ are numbers of protein and ligand atoms, respectively, N_tors_ is the number of ligand torsions, i.e., the internal rotations around ordinary chemical bonds.

Protein	PDB ID	Res, Å	N_P_	N_L_	N_tors_
GNB/LNB-binding protein	2Z8D	1.9	5897	51	6
	2Z8E	2.0	5897	51	6
α-fucosidase	2XII	1.8	7042	51	4
KIV-10 module of Apo (a)	3KIV	1.8	1206	20	5
BET protein	4MR5	1.6	1860	42	3
	4MR6	1.7	1860	49	6
CRP	1HW5	1.8	3284	33	1
Trypsin	1C5P	1.4	3220	18	1
	1K1J	2.2	3220	68	10
	2ZDM	1.9	3220	59	9
	2ZDN	2.0	3220	58	9
	2ZFS	1.5	3220	64	9
YKL-39	4P8V	1.6	5741	57	8
Factor Xia	4CRC	1.6	3711	60	11
	4CRD	2.1	3692	57	11
EngF	1J84	2.0	2642	87	10
Mp1p-LBD2	5CSD	1.5	2407	53	14
HIV-1 protease	1MRX	2.0	3140	74	11
	1MSM	2.0	3138	78	12
	2PYM	1.9	3100	86	12
	2PYN	1.9	3116	86	12
	3KDB	1.7	3138	86	13
	3NU3	1.0	3134	70	13
	4LL3	2.0	3134	75	13
Renin	2IKO	1.9	5144	46	5

**Table 2 nanomaterials-12-00274-t002:** The INN indices of minima found by the quasi-docking procedure using either PM6-D3H4X or PM7 with the COSMO solvent at the second step of the procedure. PDB ID is the identifier of the complex in Protein Data Bank, RMSD is the standard deviation between the ligand position corresponding to the global energy minimum and the crystallized ligand position. N_min_ is the number of unique energy minima found by the FLM program at the first step of the quasi-docking procedure.

PDB ID	N_min_	PM6-D3H4X COSMO		PM7 COSMO	
		INN	RMSD, Å	INN	RMSD, Å
1C5P	5349	1	0.44	1	0.43
1HW5	6848	1	0.48	1	0.48
1J84	8192	11	5.64	1	1.97
1K1J	8101	1	0.33	1	0.33
1MRX	2627	1	1.35	1	0.47
1MSM	6030	1	1.87	1	1.87
2IKO	2622	1	0.49	1	0.49
2PYM	4340	1	1.66	1	1.66
2PYN	4953	1	1.22	1	1.22
2XII	8192	1	0.58	1	0.58
2Z8D	8192	5	3.67	2	3.67
2Z8E	8192	1	1.11	1	1.11
2ZDM	5971	1	0.91	1	0.91
2ZDN	5645	8	2.12	1	0.68
2ZFS	5986	9	2.67	2	2.67
3KDB	4504	2	2.61	1	0.96
3KIV	5363	1	0.84	1	0.75
3NU3	4935	1	0.44	1	0.44
4CRC	11809	2	2.67	2	2.67
4CRD	20222	233	10.81	1	1
4LL3	5888	3	11.67	4	8.38
4MR5	5002	2	8.05	5	8.05
4MR6	4313	1	1.16	1	1.16
4P8V	8193	37	10.11	1	0.58
5CSD	29528	2576	10.33	993	10.33

**Table 3 nanomaterials-12-00274-t003:** The experimentally measured ΔH_exp_ and calculated ΔH_bind_ binding enthalpies; PDB ID is the identifier of the complex in Protein Data Bank; R is the correlation coefficient between ΔH_exp_ and ΔH_bind_.

PDB ID	ΔH_exp_, kcal/mol	ΔH_bind_, kcal/mol	
		PM6-D3H4X COSMO	PM7 COSMO
1C5P	−4.52	−27.70	−54.75
1HW5	−0.97	−37.04	−54.74
1K1J	−9.46	−29.62	−82.71
1MRX	−2.10	−9.36	−54.89
1MSM	−7.60	−20.13	−67.86
2IKO	−9.50	−33.81	−81.19
2XII	−9.80	−72.28	−92.09
2ZDM	−7.24	−30.73	−82.20
2ZDN	−5.09	–	−85.08
3KDB	−1.55	–	−54.68
3NU3	−7.30	−6.78	−54.46
4MR6	−4.04	−15.27	−47.42
R		0.4	0.74

## Data Availability

Not applicable.

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
