# Peer review of "Quantum-Chemical Quasi-Docking for Molecular Dynamics Calculations"

_nanomaterials, 2022, doi:10.3390/nano12020274_

Round 1
Reviewer 1 Report
In the paper entitled: "Quantum‐Chemical Quasi‐Docking for Molecular Dynamics Calculations" by Sulimov A. et al, the authors apply semi-empirical quantum mechanics methods to rank docking poses in order to improve the accuracy of docking predictions. To this end, they mainly focus on the difference between two quantum-mechanics methods PM6‐D3H4X and PM7. Though the paper is clearly written, there are a few issues with the methods used and with the results.
p3. The authors state that the docking is done without the solvent for the efficiency. However, on page 4 it is stated that the solvent is important. The authors should perform docking with the solvent model to show its effect on the binding poses.
p4. Non-polar contributions are not considered in this paper, which can significantly affect the results, and may explain the very strong deviation between predicted and experimental binding enthalpies in table 3.
p5 The authors should explain the choice of the pH of 7.4. Why pH values of the experimental crystal structures were not used instead?
p6 The authors mention that their optimizations reached the global minima, however, no statistical analysis of the convergence was provided.
p7 Predicted binding enthalpy strongly deviates from the experimental values, by an order of magnitude, actually larger than the difference between the results with the pm6 and pm7 semiempirical methods. The authors should explain this large deviation, and also provide individual values for the QM and implicit solvent contributions and analyze what contribution
Finally, the word "quasi" in "quasi-docking" should be explained.
Reviewer 2 Report
The manuscript ID entitled "Quantum‐Chemical Quasi‐Docking for Molecular Dynamics Calculations" is an interesting study. Molecular modeling is becoming an important tool used in the early stages of the drug development pipeline. At this stage, new inhibitors of a given therapeutic target protein are discovered. Target proteins are responsible for the progression of the disease. Inhibitors are molecules (ligands) that selectively bind to a target protein, block its function and alter the progression of the disease. New inhibitors may become drugs after their efficacy and safety have been demonstrated
in human clinical trials. Molecular modeling, mainly docking and molecular dynamics, helps to find inhibitors with a high affinity for target proteins. Docking programs perform the positioning of ligands
in a protein and estimate the protein‐ligand binding energy. Positioning of the ligand is carried out by finding the global minimum of the energy of the protein‐ligand complex. All existing docking
programs use classical force fields to describe the interaction of ligands with proteins, which is one of the reasons for the low docking accuracy and, accordingly, the low reliability of inhibitor prediction.
In this work, the quantum‐chemical quasi‐docking procedure is used to compare the docking accuracy of two quantum‐chemical semiempirical methods: PM6‐D3H4X and PM7. The latter demonstrates the best docking accuracy. I appreciate the authors for their important study in CADD. However, the following quires need to be addressed,
The authors may revise the abstract with the highlights of methods, results, and appropriate conclusions instead of more general info.
Whether the author compares their Quasi‐Docking results with classical methods? and I think, it may answer the efficiency of Quasi‐Docking instead classical method (i.e. Genetic algorithms or global energy minima)
Is there any correlation between binding enthalpy and MM-GBSA/MM-PBSA?
If the authors include figures with protein-ligand interaction and compare them with regular docking that seems more efficient.
I think the authors may improve their discussion part to highlight the importance of the study
The conclusion must be crispy instead of a lengthy summary. It should highlight the perspective and limitations of the study.
Round 2
Reviewer 1 Report
I do not think that this paper, at least in the current format is ready for the publication, and that any of my comments was properly addressed by the authors. For example, author's statement that "the faster performance without solvent allows the docking to find a more low energy minima and/or lower energy minimum" signifies that the method search does not reach the full convergence. This is in contradiction to their statement in the reply that the method reached the convergence. No support for this statement was provided, except the sentence "The time of FLM performance was selected large enough" (I am not sure what "time of performance" signifies in this context by the way). The authors should provide a comprehensive analysis of the convergence of the methods they use, which is important to demonstrate that their results are not affected by sub-optimal structures/binding poses. The analysis should be done separately for the docking poses obtained using the force field model as well as the QM minimization in the second step.
Reviewer 2 Report
Still, there are a few revisions needed as follows,
Many abbreviations are not in a full form while appearing at first on the main text ex. COSMO, MMFF94, MOPAC, COVID-19....etc. The authors need to check throughout the manuscript.
I think the author needs to enclose these statements in the discussion, which may highlight their study. " We did not calculate the correlation between experimentally measured binding enthalpy values and values of binding enthalpy calculated using MM-GBSA/MM-PBSA methods. However, we investigated docking positioning accuracy of the MM-GBSA method in several of our articles [8, 9, 10], and we showed that docking positioning accuracy with the MM-GBSA method is much worse than the docking positioning accuracy of quasi-docking with PM7+COSMO. There is no sense to calculate the binding enthalpy using the wrong ligand positions found using the MM-GBSA method".
Besides, the "MM-GBSA method" is not a wrong method, it may be less sensitive to their Q-docking. Moreover, Quantum methods are computationally bit expensive, how could be beneficial/alternative to existing ones?
The author might need to highlight their changes in discussion and conclusion
